# The Second Case of Non-Mosaic Trisomy of Chromosome 26 with Homologous Fusion 26q;26q in the Horse

**DOI:** 10.3390/ani12070803

**Published:** 2022-03-22

**Authors:** Sharmila Ghosh, Josefina Kjöllerström, Laurie Metcalfe, Stephen Reed, Rytis Juras, Terje Raudsepp

**Affiliations:** 1Department of Veterinary Integrative Biosciences, Texas A&M University, College Station, TX 77843, USA; sharmila173@gmail.com (S.G.); jkjollerstrom@cvm.tamu.edu (J.K.); rjuras@cvm.tamu.edu (R.J.); 2Rood & Riddle Equine Hospital, Lexington, KY 40580, USA; lmetcalfe@roodandriddle.com (L.M.); sreed@roodandriddle.com (S.R.)

**Keywords:** karyotyping, FISH, STR genotyping, parental origin, congenital abnormalities, neurologic disorders, Down syndrome

## Abstract

**Simple Summary:**

We present chromosome and DNA analysis of a normal Thoroughbred mare and her abnormal foal born with neurologic defects. We show that the foal has an abnormal karyotype with three copies of chromosome 26 (trisomy chr26), instead of the normal two. However, two of the three chr26 have fused, forming an unusual derivative chromosome. Chromosomes of the dam are normal, suggesting that the chromosome abnormality found in the foal happened during egg or sperm formation or after fertilization. Analysis of the foal and the dam with chr26 DNA markers indicates that the extra chr26 in the foal is likely of maternal origin and that the unusual derivative chromosome resulted from the fusion of two parental chr26. We demonstrate that although conventional karyotype analysis can accurately identify chromosome abnormalities, determining the mechanism and parental origin of these abnormalities requires DNA analysis. Most curiously, this is the second case of trisomy chr26 with unusual derivative chromosome in the horse, whereas all other equine trisomies have three separate copies of the chromosome involved. Because horse chr26 shares genetic similarity with human chr21, which trisomy causes Down syndrome, common features between trisomies of horse chr26 and human chr21 are discussed.

**Abstract:**

We present cytogenetic and genotyping analysis of a Thoroughbred foal with congenital neurologic disorders and its phenotypically normal dam. We show that the foal has non-mosaic trisomy for chromosome 26 (ECA26) but normal 2n = 64 diploid number because two copies of ECA26 form a metacentric derivative chromosome der(26q;26q). The dam has normal 64,XX karyotype indicating that der(26q;26q) in the foal originates from errors in parental meiosis or post-fertilization events. Genotyping ECA26 microsatellites in the foal and its dam suggests that trisomy ECA26 is likely of maternal origin and that der(26q;26q) resulted from Robertsonian fusion. We demonstrate that conventional and molecular cytogenetic approaches can accurately identify aneuploidy with a derivative chromosome but determining the mechanism and parental origin of the rearrangement requires genotyping with chromosome-specific polymorphic markers. Most curiously, this is the second case of trisomy ECA26 with der(26q;26q) in the horse, whereas all other equine autosomal trisomies are ‘traditional’ with three separate chromosomes. We discuss possible ECA26 instability as a contributing factor for the aberration and likely ECA26-specific genetic effects on the clinical phenotype. Finally, because ECA26 shares evolutionary homology with human chromosome 21, which trisomy causes Down syndrome, cytogenetic, molecular, and phenotypic similarities between trisomies ECA26 and HSA21 are discussed.

## 1. Introduction

Multiple forms of chromosome rearrangements have been reported in the domestic horse, *Equus caballus* (ECA) and most are associated with decreased fertility, embryonic or fetal loss, congenital and developmental disorders, causing significant economic loss to breeders and the equine industry [1,2]. The most commonly found chromosomal abnormalities in horses are X-monosomy and XY male-to-female sex reversal (also known as XY disorder of sex development or XY DSD) [1,2,3], which owe to the specific features of equine sex chromosome organization [2,4,5]. Rearrangements involving autosomes, however, are rare in horses and include mainly a few translocations and autosomal aneuploidies [2].

Aneuploidies cause genetic imbalance, due to which most of them are lethal [6], and the 14 reported live-born cases of autosomal trisomies involve only the six smallest equine autosomes—ECA23, 26, 27, 28, 30 and 31 [1,2,7]. Autosomal aneuploidies are equally rare in other domestic species. There are 16 reported cases of autosomal trisomies in cattle involving the 10 smallest autosomes, typically resulting in fetal death or postnatal culling by breeders due to congenital defects [8,9]. In the domestic pig, there are no reports of live-born animals with whole autosome aneuploidies [10], and all autosomal trisomies in dogs have exclusively been found in tumor cells [11]. Likewise, although aneuploidies occur in at least 5% of clinically recognized human (*Homo sapiens*, HSA) pregnancies and account for over 25% of spontaneous abortions, only trisomies of HSA13, 18 and 21 have been found in live born, of which only trisomy HSA21 survives to adulthood [12,13].

Extensive studies of human autosomal aneuploidies show that the majority are caused by errors in maternal meiosis I (MI) with advanced maternal age being a critical contributing factor, whereas only 5–10% of trisomies are caused by paternal errors [13]. At the same time, human data also show remarkable variation among trisomies regarding the parent and meiotic stage (MI or MII) of origin of the extra chromosome. For example, paternal errors account for nearly 50% of trisomy HSA2 but almost never for trisomy HSA16. Likewise, errors in maternal MI account for almost all cases of trisomy HSA16, whereas trisomy HSA18 is predominantly caused by errors in maternal MII, suggesting that the patterns of non-disjunction may have chromosome specific effects [13,14]. 

In rare occasions, trisomies of acrocentric autosomes are combined with Robertsonian fusion or isochromosome formation [15,16,17], so that despite of aneuploidy, the diploid chromosome number remains normal. For example, about 5–6% of cases with Down syndrome carry unbalanced heterologous or homologous fusions involving HSA21 [15,17]. The mechanism for heterologous fusions is Robertsonian translocation, of which the most common (82%) in Down syndrome patients is rob (14q;21q), with the remaining 8% represented by rob(13q;21q), rob(15q;21q) and rob(21q;22q) [17]. On the other hand, trisomy due to homologous fusion of (21q;21q) can result from different mechanisms—by isochromosome i(21q) formation or due to Robertsonian translocation rob(21q;21q). Since isochromosomes result from the duplication of a single chromosome arm [18], the duplicated parts are genetically identical and can be distinguished from homologous translocation by genotyping for allelic variation using chromosome specific polymorphic short tandem repeat (STR) markers [15,16,18,19]. 

In domestic animals, the only case of autosomal trisomy combined with centric fusion or isochromosome formation has been reported in horses for trisomy ECA26 [20,21]. The karyotype formula of the affected Thoroughbred mare was presented as 64,XX, −26,+t(26q;26q), but because polymorphic STR markers were not available for horses at that time, the researchers could not determine whether the abnormal chromosome (26q;26q) was an isochromosome or the result of a Robertsonian fusion. 

In the present study, we report and characterize the second equine case of trisomy 26 involving homologous fusion 26q;26q. We will characterize the case using classical and molecular cytogenetic approaches and genotype the affected individual and its dam with ECA26 STR markers to determine the mechanism and likely parental origin of the aberration. 

## 2. Material and Methods

### 2.1. Ethics Statement

Procurement of blood samples followed the United States Government Principles for the Utilization and Care of Vertebrate Animals Used in Testing, Research and Training. These protocols were approved as AUP and CRRC #2018-0342 CA at Texas A&M University.

### 2.2. Case Description and Sampling

A Thoroughbred foal (ID: H1063) was euthanized at the age of 5 months and 3 weeks due to stupors that gradually developed into ataxia, due to failure to thrive despite nursing well and being initially treated for possible neonatal mal-adjustment syndrome, and due to being inappropriate mentally. Although cervical radiographs did not provide an explanation for progressing ataxia, necropsy revealed axonal degeneration in brainstem and spinal cord suggestive of equine degenerative myeloencephalopathy. This was the first foal of a 5-year-old maiden Thoroughbred mare boarded on a large, well-managed farm. The sire had had several normal foals before. Peripheral blood samples in EDTA- and sodium heparin-containing vacutainers (VACUTAINERTM, Becton Dickinson) were obtained from the affected foal and its dam (ID: H1066) for cytogenetic and DNA analysis.

### 2.3. Cell Cultures and Chromosome Preparations

Metaphase chromosome spreads were prepared from peripheral blood lymphocytes following standard protocols [22]. Briefly, 1 mL of sodium heparin stabilized peripheral blood was grown for 72 h in 9 mL of culture medium RPMI-1640 supplemented with HEPES and Glutamax (Gibco), 30% fetal bovine serum (FBS; R&D Systems Inc., Minneapolis, MN, USA), 1X antibiotic-antimycotic (100×; Invitrogen, Waltham, MA, USA), and 15 µg/mL pokeweed mitogen (Sigma Aldrich, St. Louis, MO, USA). Lymphocyte cultures were harvested with demecolcine solution (10 µg/mL; Sigma Aldrich), treated with Optimal Hypotonic Solution (Rainbow Scientific, Windsor, CT, USA), and fixed in 3:1 methanol/acetic acid. The cells were dropped on clean, wet glass slides and checked under phase contrast microscope (×200) for quality. 

### 2.4. Karyotyping and Cytogenetic Analysis

Chromosomes were stained by GTG-banding [23] for karyotyping. Karyotyping and chromosome analysis were performed with a motorized fluorescence microscope Axio Imager M2p (Zeiss) equipped with a high-resolution progressive scan CCD camera CoolCube 1 and Ikaros v5.3.18 software (MetaSystems GmbH, Altlußheim, Germany). Images of a minimum of 30 cells were captured and analyzed per individual. Horse chromosomes were identified and arranged into karyotypes according to the International System of Cytogenetic Nomenclature of the Domestic Horse [24] and chromosome aberrations were described following Human Cytogenomic, Nomenclature [25].

### 2.5. Fluorescence In Situ Hybridization (FISH)

The rearrangements identified by conventional cytogenetic analysis were validated by two-color FISH with ECA26-specific Bacterial Artificial Chromosome (BAC) clones (Table 1) from horse genomic BAC library CHORI-241 (https://bacpacresources.org/, last accessed 1 December 2021). The probes were labeled with biotin or digoxigenin by nick translation using Biotin or DIG Nick Translation Mix (Roche Diagnostics, Basel, Switzerland), following the manufacturer’s protocol. Hybridization and signal detection followed standard protocols described elsewhere [22]. Biotin-labeled probes were detected with Alexa Fluor^®^ 488 streptavidin conjugate (Molecular Probes, Life Technologies, Carlsbad, CA, USA) and digoxigenin-labeled probes with DyLight^®^ 594 anti-digoxigenin conjugate (Vector Laboratories, Burlingame, CA, USA). Chromosomes were counterstained with 4′,6-diamidino-2-phenylindole (DAPI). At least 10 cells were captured and analyzed for each experiment using Isis v5.3.18 software (MetaSystems GmbH, Altlußheim, Germany).

### 2.6. DNA Isolation, PCR Analysis and STR Genotyping

Genomic DNA was isolated from EDTA-stabilized blood with QIAamp DNA Blood Mini Kit (Qiagen, Hilden, Germany). Both horses were tested by PCR for the Y-linked *SRY* gene and X-linked androgen receptor (*AR*) gene as described earlier [27], followed by genotyping for the 15 autosomal STRs of the standard equine parentage panel [28], and an additional 24 STRs specific for ECA26 (Table 2). Genotyping was performed either with directly fluorescently labeled primers [29] or with three-primer nested PCR where the forward primer in each primer-pair had an M13-tail which was targeted by a fluorescently labeled universal M13 primer during PCR reactions [30]. Annealing temperature for all PCR reactions was 58 °C. The PCR products were resolved with an ABI PRISM 377 (Applied Biosystems, Foster City, CA, USA) and allele sizes were determined using GeneScan-500 LIZ Size Standard and GeneMapper^®^ v4.1 (Applied Biosystems, Waltham, MA, USA). 

## 3. Results

### 3.1. Chromosome Analysis

Cytogenetic analysis showed that the affected foal (H1063) had normal 2n = 64 diploid number, XY sex chromosomes, one copy of normal ECA26, and the karyotype contained a morphologically abnormal metacentric derivative chromosome (Figure 1A,B). Analysis of GTG-banding suggested that the derivative chromosome was composed of two copies of ECA26 likely fused at the centromeres. Molecular cytogenetic analysis by FISH with two ECA26 BAC clones, one corresponding to the proximal (BAC 9N4) and the other, to the distal (BAC 91H11) portion of the chromosome, confirmed that the derivative chromosome was the result of homologous centric fusion 26q;26q (Figure 1C). Thus, despite the normal diploid number, the foal carried trisomy ECA26 in all cells analyzed. However, by cytogenetic analysis alone, it was not possible to determine whether the derivative chromosome resulted from Robertsonian fusion rob(26q;26q) or from isochromosome formation i(26q).

Karyotype analysis of the dam (H1066) of the abnormal foal showed normal 64,XX female karyotype (Figure 1D,E) indicating that chromosomal abnormality of the foal must have originated from a parental meiotic error or a post-fertilization zygotic event. 

As a standard part of cytogenetic analysis, both horses were tested by PCR for the *SRY* and *AR* genes and the results agreed with karyotype analysis and the phenotypic sex of the two horses: the XY foal H1063 was *SRY*-positive, the XX dam H1066 was *SRY*-negative, and both horses were positive for the X-linked control marker *AR*.

### 3.2. STR Genotyping: Parentage and the Origin of ECA26 Trisomy

Genotyping for 15 genome-wide autosomal STRs [28] qualified the cytogenetically normal Thoroughbred mare H1066 as the dam of the affected foal H1063. The two horses were also genotyped for 24 STR markers which were evenly distributed over ECA26, starting with UMNe588 as the most proximal marker and ending with TKY523 as the most distal one (Table 3). As expected, the STR markers showed the presence of one or two alleles in the cytogenetically normal dam H1066. However, five STRs had three alleles in the abnormal foal H1063 (Figure 2, Table 3), indicating that the metacentric derivative chromosome was the result of Robertsonian fusion rob(26q;26q) and not an isochromosome. The karyotype of the foal was designated as 64,XY,der(26),rob(26q;26q) [25].

Further comparison of the genotyping patterns between the foal and the dam showed that in all 5 cases where the foal had 3 alleles, two of the alleles were identical with those of the dam (Figure 2). Additionally, of the 10 markers that were heterozygous both in the foal and the dam, the two horses shared the same alleles (Table 3). Based on these observations, and despite having no genotype information for the sire, it is very likely that the extra ECA26 in the foal was of maternal origin.

## 4. Discussion

Here, we characterized by chromosome analysis and STR genotyping an equine case of trisomy for chromosome 26 with homologous fusion 26q;26q (Figure 1 and Figure 2, Table 2). Genotyping ECA26 STRs in the affected horse and its dam showed that the abnormal chromosome was the result of Robertsonian translocation and most likely of maternal origin. Since the dam of the affected foal had normal 64,XX karyotype (Figure 1D), the aneuploidy must have originated from maternal meiotic nondisjunction, though the following fusion could have taken place either in meiosis or post-fertilization.

It is certainly curious that this is the second case of trisomy ECA26 with a derivative chromosome 26q;26q in horses. The first case was described more than three decades ago [20,21], but because of uninformative blood typing, the mechanism (Robertsonian fusion or isochromosome) or parental origin of the aneuploidy remained unknown [21]. In our case, the presence of three alleles for 5 ECA26 STRs in the affected foal (Figure 2, Table 2) was a compelling piece of evidence that the derivative metacentric chromosome resulted from Robertsonian fusion. Furthermore, since all heterozygous STRs of the dam had the same two alleles also present in the affected foal (Table 2), we concluded that the extra chromosome ECA26 was likely of maternal origin. Though, complete evidence for the parental origin requires STR genotyping of the sire, whose samples were not available. Nevertheless, the findings underscore the importance of combining STR genotyping with cytogenetic analysis of possible isochromosomes or Robertsonian fusions. Isochromosome is formed by centromere mis-division of sister chromatids resulting in a bi-armed chromosome with identical genetic material in each arm [18,32]. Homologous Robertsonian fusions, on the other hand, result in genetically distinct arms preserving the heterozygosity from the parent from which the extra chromosome came from [16,18,33].

Another intriguing aspect of the present and the previous case [21] was that there have been no reports about ECA26 trisomy with three separate copies of the chromosome. This contrasts with other recurrently reported equine trisomies: all cases of cytogenetically studied trisomies of ECA27 (4 cases), ECA30 (5 cases), and ECA31 (2 cases) (reviewed by [2]) involve three separate chromosomes without homologous fusions. Furthermore, the trisomy ECA26 described in this study, is so far the only confirmed Robertsonian fusion in equine clinical cytogenetics [2], even though Robertsonian type rearrangements have been a normal part of equid and Perissodactyl karyotype evolution [34].

Can it be that ECA26 is more prone for centric fusion than other equine small acrocentric chromosomes? Chromosome-specific effects have been observed in humans where a small percentage of cases of Down and Patau syndrome with trisomy HSA21 and HSA13, respectively, have the extra chromosome in the form of Robertsonian fusion or an isochromosome [15,16,17,35,36]. In Down syndrome, there are even rare mosaic cases where one cell line carries HSA21 isochromosome and another, a Robertsonian fusion [17]. It is thought that some human chromosomes, such as HSA21, are inherently unstable and more prone to rearrangements [17] due to certain features of their sequence architecture (e.g., region-specific low copy number repeats) [18]. Based on our current knowledge of the horse genome [37], ECA26 does not stand out with any sequence peculiarities. Additionally, unlike HSA21 and other human acrocentric autosomes, ECA26 does not carry the satellite with multicopy rRNA genes that may contribute to instability [18]. On the other hand, and based on comparative chromosome painting [38] and gene mapping [26], ECA26 is more similar to HSA21 than to any other human chromosome because about 30 Mb (70%) of ECA26 shares evolutionary homology with the entire HSA21. However, the remaining 13 Mb (30%) of ECA26 is homologous to a part of HSA3 and this happens to be the pericentromeric/proximal portion of ECA26 which is involved in homologous fusion 26q;26q. Therefore, it is perhaps not relevant to expand the known instability of HSA21 [17] to ECA26 and it remains unclear whether the two cases of ECA26 trisomy with 26q;26q fusion were merely a coincidence or true reflections of presently unknown sequence properties of this horse autosome.

On the other hand, it is also possible that ECA26 instability and rearrangements are due to sequence variants segregating in certain horse breeds or families and not due to the genomic architecture of ECA26 per se. Indeed, the case described in this study and the one reported earlier [20,21], both occurred in Thoroughbreds. However, then again, two cases are too few for any conclusions.

Besides cytogenetics, there are several other shared features of interest between the two cases of trisomy ECA26 (this study; [20,21]). In both, the dams of the affected foals were young—5 years-old in this case and 3 years-old in the one described by Bowling et al. [21], thus excluding advanced maternal age as a contributing factor and rather supporting chromosome-specific effects. Additionally, both affected horses had gait deficits (ataxia), were not thriving, and had behavioral and mental issues. However, because the case presented in this study resulted in euthanasia at a young age but the horse described by Bowling et al. [21] lived many years, the basis of comparison is rather limited. It is, though, noteworthy that necropsy of the present case showed axonal degeneration in brainstem and spinal cord as seen in equine degenerative myeloencephalopathy (EDM) [39]. Although genetic basis for EDM is suspected but currently unknown [39], the present findings suggest that possible contribution of chromosome abnormalities/genetic imbalance should be considered. The fact that both cases were described as “inappropriate mentally” (this study) or “mentally dull” [21], and because of the homology between ECA26 and HSA21, there is a temptation to compare equine trisomy 26 with human Down syndrome. Indeed, there are some similarities: the horse described by Bowling et al. [21] lived many years and it is well-known that trisomy HSA21 is the only human autosomal trisomy surviving to adulthood [12,13]. Furthermore, at the age of 4, the mare with trisomy ECA26 gave birth to a chromosomally normal colt [21], and there are many cases of fertile women with Down syndrome in humans [40]. Despite this, drawing parallels between the two cases of ECA26 trisomy in horses with human Down syndrome should be taken with great caution. Firstly, genetic homology between ECA26 and HSA21 is not one-to-one since ECA26 is homologous also to part of HSA3 [26,38]. Secondly, stupors and ataxia which were the prevailing features of the two equine cases, are not the predominant characteristics of Down syndrome [41]. Most importantly, however, it is extremely narrow to compare the few phenotypic characteristics of two equine cases with the extensive research and clinical material available for Down syndrome since 1866 [41]. Furthermore, phenotypic features of the two equine cases share similarities with the phenotypes of other reported equine autosomal aneuploidies. For example, gait deficiencies, behavioral abnormalities and poor thriving have also been found in cases of trisomy ECA27 and ECA30 (reviewed by [2]), thus not being unique to trisomy ECA26. All in all, it is hard to tell which phenotypic features of trisomy ECA26 are the specific consequences of ECA26 overdose and which ones are due to general genomic imbalance.

## 5. Conclusions

We demonstrated that proper characterization of an autosomal (ECA26) trisomy with homologous fusion (26q;26q) and determining the mechanism and parental origin of the rearrangement, require the use of complementary approaches—cytogenetics and genotyping. To date, equine trisomy with homologous fusion has been unique to ECA26. However, to determine whether this is an ECA26-specific effect or just a coincidence, requires more cytogenetic cases and improved knowledge about the genomic architecture and functional annotation of ECA26. The latter is also needed to shed more light on the possible homology between trisomy ECA26 in the horse and the Down syndrome with trisomy HSA21 in humans.

## Figures and Tables

**Figure 1 animals-12-00803-f001:**
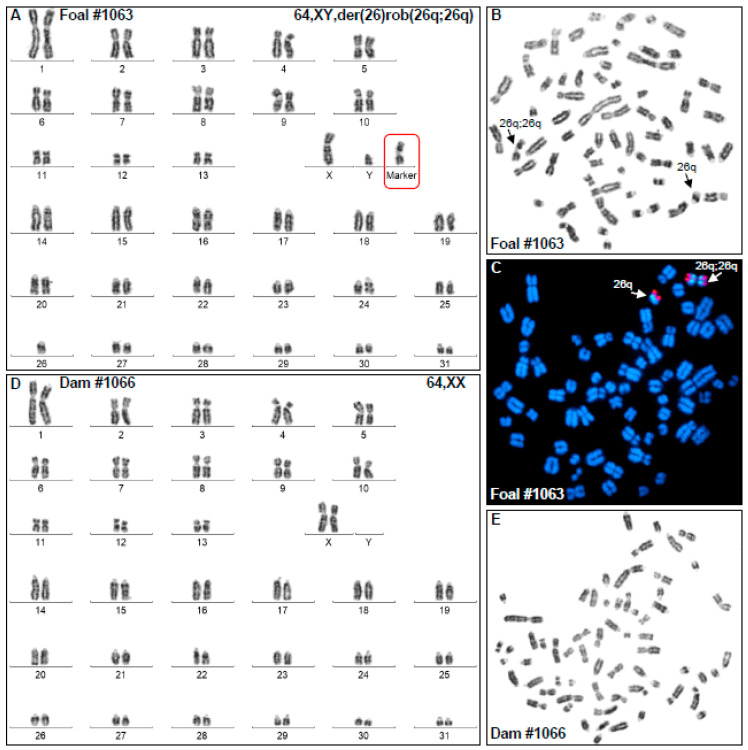
Cytogenetic analysis results. (**A**) GTG-banded karyotype of the affected foal H1063 showing 64,XY karyotype with a single ECA26 and a metacentric derivative chromosome with the arms corresponding to ECA26q; (**B**) Metaphase spread corresponding to H1063 karyotype; arrows show the normal and derivative ECA26; (**C**) FISH results with ECA26 BAC clones (BAC 9N4 green; BAC 91H11 red) showing (arrows) the presence of a single ECA26 and a metacentric derivative chromosome 26q;26q; (**D**) GTG-banded karyotype of the dam (H1066) showing normal 64,XX female karyotype; (**E**) Metaphase spread corresponding to the karyotype of the dam (H1066).

**Figure 2 animals-12-00803-f002:**
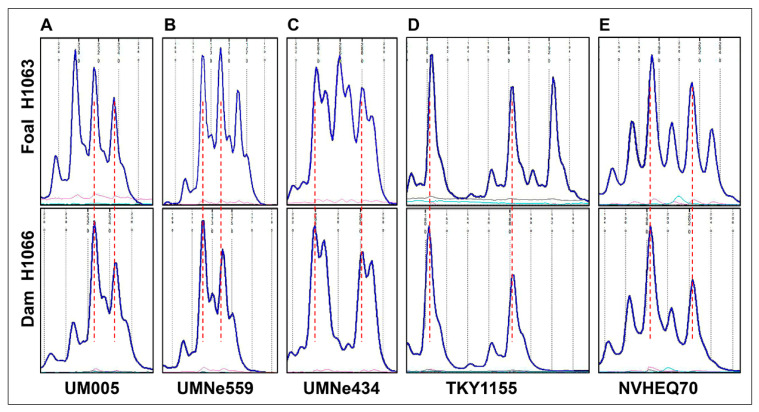
Genotyping results for ECA26 STRs UM005 (**A**), UMNe559 (**B**), UMNe434 (**C**), TKY1155 (**D**), and NVHEQ70 (**E**) showing the presence of three alleles in the foal H1063 (upper row) and two alleles in its dam H1066 (lower row). Note that for each STR, the two alleles present in the dam are shared with the foal. Allele size scales are aligned between the foal and the dam (vertical red dotted lines).

**Table 1 animals-12-00803-t001:** Information about ECA26 BAC clones used for FISH. Genomic location of BACs was retrieved from NCBI Genome (https://www.ncbi.nlm.nih.gov/genome/, last accessed 15 October 2021) and cytogenetic map information from [26].

CHORI-241 BAC Clone	BAC Location in EquCab3	Cytogenetic Location	Representative Genes
9N4	chr26:12,142,705–12,318,937	26q14	*ROBO2*
91H11	chr26:42,857,954–43,065,765	26q17	*S100B*

**Table 2 animals-12-00803-t002:** Information about ECA26 STR markers used for genotyping.

STR	Forward Primer: 5′-3′	Reverse Primer: 5′-3′	NCBI Accession or Reference
A-17 **	GTGGAGAGATAAAAGAAGATCC	GGCCACAAGGAATGAACACAC	X94446
COR071 **	CTTGGGCTACAACAGGGAATA	CTGCTATTTCAAACACTTGGA	AF142608
LEX044 *	TTGGGCTTCTTATCTTGTTAC	GGCCATATGATTTGCTTT	AF075646
NVHEQ070 **	GCTGGTCAAGTCACACTGTG	AACCTCACCCCAAGTTGTAT	AJ245765
TKY1155 *	AGCTCAGGGCGAATCTTACA	AAACCTGGGCATCTTCCTTT	AB104373
TKY275 *	TCTCAGTGGATATAACTAGC	GAGATGGATACAGATAGAAG	AB033926
TKY3385 *	TGACACCACCAGGGAAAAGT	CATGTTCCCTCACCTCTGGT	AB217328
TKY414 *	CCTGAAATCCGCTTCCATTA	ACCGGGTTATTTTGACATGG	AB103632
TKY488 *	TGTGTTTGTGTGCTATATACATGCTT	TGACATGAAGGCTGGACTTG	AB103706
TKY502 *	ACGGAAAACGTATGCCACTC	AGTGGGGACTTTGTTGAGGA	AB103720
TKY523 *	TGCACACCCATTCTAGCTCA	GTGGCTCACTCCTCGCTTAC	AB103741
TKY664 *	TACTGCCCTTGGCTGACTCT	CAGAACATGAACCCCTCCAG	AB103882
TKY766 *	ACTTTGCACCTGTGCAAAAAG	CTGATTCTTGGCATCTGGAAA	AB103984
TKY778 *	CTTAGATGGAGTCCTCCTAC	GGGTTCCTTTTACCTTCTCC	AB103996
TKY846 *	TCAAACCATCTGCTCAGAAG	AAATCCCAATCTGAGGGTAG	AB104064
TKY934 *	TTCCAGTGGTTAGGATGTAG	TTGAGCATAGTGATAGCATATG	AB104152
UM005 *	CCCTACCTGAAATGAGAATTG	GGCAAAAGATCAGGCCAT	AF195127
UMNe127 *	TTATAAATCACCACTGTTTACACAC	TCTTGAAGCAGGATGGGC	AY391298
UMNe153 *	GTGCTGGAGTGAGCTGACC	ATCCAAATCGGAGACCATATG	AF536265
UMNe188 *	GTTAACAAGGATTGTTTTGGGC	TGCGTTTCTGCTTCTCCC	AY391317
UMNe434 *	TCTGCTGTTGGCCATCATC	ACCTGCCTGCAAAACCTTC	[26]
UMNe542 *	TGAAAGAGACCATACACGATGC	CACGACTTAGAGACGTGTGAGC	AY735263
UMNe559 *	CTTCCCATTCTCTATCACCCC	CTGTTCTCCCAATTCTTTCTGG	[26]
UMNe588 *	CGCAGGTAGACTGTGTTAGGC	CAAGACTGGAAATTTTCAAGGG	[26]

* Forward primers had a M13 tail: TGTAAAACGACGGCCAGT ** Directly fluorescently labeled primers; Primer sequences were retrieved from [26,31].

**Table 3 animals-12-00803-t003:** Genotyping results with ECA26 STRs. Markers are presented according to their linear order from centromere to telomere in ECA26; markers with three alleles in the foal are highlighted.

ECA26 Genomic Location, EquCab3	ECA26 STR	H1063: Alleles	H1066: Alleles
5,190,320–5,190,461	UMNe588	156	156
6,518,546–6,518,920	TKY934	158/160	158/160
7,006,025–7,006,186	UMNe559	173/175/177	173/175
8,845,111–8,845,452	TKY846	201/203	201
11,835,911–11,836,148	TKY766	104/110	110
19,109,482–19,110,003	TKY502	220	220
19,136,880–19,137,134	UMNe153	142/162	142/162
19,767,544–19,767,787	COR071	202/210	202/210
20,212,459–20,212,887	TKY275	142/158	142/158
20,367,221–20,367,742	LEX044	204/218	204/218
21,795,871–21,795,973	A-17	107/109	107/109
23,979,076–23,979,467	TKY778	226	226
24,637,783–24,638,172	TKY488	107/109	107/109
26,379,056–26,379,415	UMNe127	148	148
26,766,980–26,767,353	UM005	230/232/234	232/234
31,041,466–31,041,914	TKY1155	180/188/192	180/188
31,486,888–31,487,451	NVHEQ70	198/202/204	198/202
32,006,987–32,007,419	UMNe188	142/144	142/144
34,426,999–34,427,199	TKY3385	204	204
36,846,956–36,847,298	TKY664	271	271
37,488,847–37,489,215	UMNe542	270/276	270/276
38,794,949–38,795,212	UMNe434	284/286/288	284/288
39,259,334–39,259,638	TKY414	171/173	171/173
39,552,914–39,553,412	TKY523	162	162

## Data Availability

Not applicable.

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
