# Peer review of "The Second Case of Non-Mosaic Trisomy of Chromosome 26 with Homologous Fusion 26q;26q in the Horse"

_animals, 2022, doi:10.3390/ani12070803_

Round 1

Reviewer 1 Report

The manuscript from Sharmila Ghosh et al describes an interesting case report of a Thoroughbred foal with a Robertsonian fusion causing the trisomy of chromosome 26. The authors applied a combination of cytogenetic and molecular techniques to identify the maternal origin of the abnormality. Interestingly, horse chromosome 26 is orthologous to human chromosome 21, whose trisomy is responsible for Down syndrome. The experiments are well described, the Figure of high quality and the manuscript is well written.

Additional comments:

  1. What is the main question addressed by the research?

To test whether the neurological abnormalities of a Thoroughbred foal are related to an abnormal karyotype.

  1. Do you consider the topic original or relevant in the field, and if so, why?

The topic is relevant in the field of horse breeding and genetics. In particular, a trisomy of chromosome 26 was demonstrated in the foal. Horse chromosome 26 is orthologous to human chromosome 21, whose trisomy causes Down syndrome.   

  1. What does it add to the subject area compared with other published material?

In my knowledge, this is the first example, in the horse, of a trisomy due to the presence of a marker chromosome derived from Robertsonian fusion.

  1. What specific improvements could the authors consider regarding the methodology?

The methodology is adequate

  1. Are the conclusions consistent with the evidence and arguments presented and do they address the main question posed?

Yes

  1. Are the references appropriate?

Yes

  1. Please include any additional comments on the tables and figures.

Tables are adequate and figures of high quality, particularly Figure 1.

Author Response

Thank you!

Reviewer 2 Report

In horses, chromosome rearrangements are strongly associated with fertility as well as foals or embryonic losses. Thus, the described case is interesting from the point of view of basic and applied research - future diagnostics. In my opinion, the presented care report is a valuable study focused on identifying chromosome aberration in horses. The authors used the appropriate design methods, which are described in detail. The manuscript is well written and contains all necessary information. The result is presented clearly and in a correct manner.  I recommend presented manuscript for publication.

Author Response

Thank you!

Reviewer 3 Report

In the manuscript, the authors reported a case of trisomy ECA26 with 2 homologous fusion 26q;26q in a male Thoroughbred foal. The authors proved that the chromosome rearrangement resulted from Robertsonian fusion by cytogenetic and genotyping analysis, and made a meaningful comparison of the rearrangement of ECA26 and HSA21. I have some suggestions as follows, 1. Full phrase of abbreviation “HSA” should be provided, just as the authors did to “ECA”. 2. Please explain why some forward primers had an M13-tail. Is it for the convenience for sequencing? It seems that some STRs were genotyped by sequencing. The authors should describe it clearly in the method part. 3. The annealing temperature of the 24 STRs specific for ECA26 should be provided in Table 2. 4. Line 229-232, the sentence, “However, even if we had ECA26 genotypes from both parents, we still could not determine the chromosomal origin of different ECA26 alleles, i.e., whether the arms of the derivative chromosome 26q;26q originated from one or from both parents”, may lead to misunderstanding, and It should be deleted. I can understand that it is sometimes difficult or inconvenient to get the blood sample of a sire, but it is certainly that the cytogenetic and genotyping analysis of the sire could provide more useful information to the case. 5. The conclusions should be brief and avoid repeating the content of discussion part. 6. I wonder whether there are some important functional genes in the regions where the rearrangement of ECA26 occurred. It would be interesting to discuss about it. Overall, this is an interesting case report, and the manuscript is well-organized and well-written. So I recommend the manuscript to be accepted for publication.

Author Response

We thank the Reviewer for excellent suggestions. Please see below our responses to each.

In the manuscript, the authors reported a case of trisomy ECA26 with 2 homologous fusion 26q;26q in a male Thoroughbred foal. The authors proved that the chromosome rearrangement resulted from Robertsonian fusion by cytogenetic and genotyping analysis, and made a meaningful comparison of the rearrangement of ECA26 and HSA21. I have some suggestions as follows,

  1. Full phrase of abbreviation “HSA” should be provided, just as the authors did to “ECA”.

Added Homo sapiens, HSA to Introduction, line 72.

  1. Please explain why some forward primers had an M13-tail. Is it for the convenience for sequencing? It seems that some STRs were genotyped by sequencing. The authors should describe it clearly in the method part.

Both genotyping methods have been known and widely used for genotyping (not sequencing) for decades and we have provided the text with appropriate references for methodological details. The main difference is economy – the three-primer method is much cheaper and is usually used for specific one-time projects while directly fluorescently labeled primers are typically part of regularly used parentage kits. Nevertheless, we expanded the text on lines 172-175 about genotyping as follows:

“Genotyping was done either with directly fluorescently labeled primers [29] or with three-primer nested PCR where the forward primer in each primer-pair had an M13-tail which was targeted by a fluorescently-labeled universal M13 primer during PCR reactions [30]. Annealing temperature for all PCR reactions was 58 oC. The PCR products were resolved with an ABI PRISM 377 (Applied Biosystems, Foster City, CA, USA) and allele sizes were determined using GeneScan-500 LIZ Size Standard and GeneMapper® v4.1 (Applied Biosystems, Waltham, MA, USA). “

  1. The annealing temperature of the 24 STRs specific for ECA26 should be provided in Table 2.

Please see response to the previous comment and line 173-174 in the revised manuscript.

  1. Line 229-232, the sentence, “However, even if we had ECA26 genotypes from both parents, we still could not determine the chromosomal origin of different ECA26 alleles, i.e., whether the arms of the derivative chromosome 26q;26q originated from one or from both parents”, may lead to misunderstanding, and It should be deleted. I can understand that it is sometimes difficult or inconvenient to get the blood sample of a sire, but it is certainly that the cytogenetic and genotyping analysis of the sire could provide more useful information to the case.

The sentence is deleted as per Reviewer’s suggestion.

  1. The conclusions should be brief and avoid repeating the content of discussion part.

Thank you. We revised and shortened Conclusions. It reads now as follows:

  1. Conclusions

“We demonstrated that proper characterization of an autosomal (ECA26) trisomy with homologous fusion (26q;26q) and determining the mechanism and parental origin of the rearrangement, require the use of complementary approaches - cytogenetics and genotyping. To date, equine trisomy with homologous fusion has been unique to ECA26. However, to determine whether this is an ECA26-specific effect or just a coincidence, requires more cytogenetic cases and improved knowledge about the genomic architecture, and functional annotation of ECA26. The latter is also needed to shed more light on the possible homology between trisomy ECA26 in the horse and the Down syndrome with trisomy HSA21 in humans.”

  1. I wonder whether there are some important functional genes in the regions where the rearrangement of ECA26 occurred. It would be interesting to discuss about it. Overall, this is an interesting case report, and the manuscript is well-organized and well-written. So I recommend the manuscript to be accepted for publication.

ECA26 is homologous to the entire HSA21 and a small part of HSA3 and, thus, carries over 1200 orthologous protein coding genes. It is not possible to specifically pinpoint any. No changes made.

Reviewer 4 Report

Dear authors,

The manuscript is very well conducted, presented, executed and written.

The information herein presented contribute to cytogenetic and chromosomal rearrangements studies in livestock. The theme has been low studied in livestock since molecular and genomic studies started. It is quite important to keep on doing researches like this.

Moreover, in this particular case, there are some applications even in human genetics.

I recommend the manuscript for publication.

Author Response

Thank you!